

# Preference of a native beetle for "exoticism," characteristics that contribute to invasive success of *Costelytra zealandica* (Scarabaeidae: Melolonthinae)

Marie-Caroline Lefort[1,2], Stéphane Boyer[2], Jessica Vereijssen[3], Rowan Sprague[1], Travis R. Glare[1] and Susan P. Worner[1]

[1] Bio-Protection Research Centre, Lincoln, New Zealand
[2] Department of Natural Sciences, Unitec Institute of Technology, Auckland, New Zealand
[3] The New Zealand Institute for Plant & Food Research Limited, Lincoln, New Zealand

Corresponding author
Marie-Caroline Lefort, Marie-Caroline.Lefort@lincolnuni.ac.nz

## ABSTRACT

Widespread replacement of native ecosystems by productive land sometimes results in the outbreak of a native species. In New Zealand, the introduction of exotic pastoral plants has resulted in diet alteration of the native coleopteran species, *Costelytra zealandica* (White) (Scarabaeidae) such that this insect has reached the status of pest. In contrast, *C. brunneum* (Broun), a congeneric species, has not developed such a relationship with these 'novel' host plants. This study investigated the feeding preferences and fitness performance of these two closely related scarab beetles to increase fundamental knowledge about the mechanisms responsible for the development of invasive characteristics in native insects. To this end, the feeding preference of third instar larvae of both *Costelytra* species was investigated using an olfactometer device, and the survival and larval growth of the invasive species *C. zealandica* were compared on native and exotic host plants. *Costelytra zealandica*, when sampled from exotic pastures, was unable to fully utilise its ancestral native host and showed higher feeding preference and performance on exotic plants. In contrast, *C. zealandica* sampled from native grasslands did not perform significantly better on either host and showed similar feeding preferences to *C. brunneum*, which exhibited no feeding preference. This study suggests the possibility of strong intraspecific variation in the ability of *C. zealandica* to exploit native or exotic plants, supporting the hypothesis that such ability underpins the existence of distinct host-races in this species.

## INTRODUCTION

By widely replacing native ecosystems with more economically productive land, modern intensive agriculture has often been regarded by ecologists as a driver for substantial biodiversity loss (*Robinson & Sutherland, 2002*; *Tilman et al., 2002*; *Foley et al., 2005*). Although detrimental for numerous species, anthropogenic modifications creating novel

ecological conditions appear to be beneficial under certain circumstances for some native species. For instance, it is acknowledged that the high diversity of phytophagous insects partially results from evolutionary processes that occur through the action of factors affecting their diet breadth (*Gaete-Eastman et al., 2004*), such as the appearance of a new host plant. Hence, the ecological repercussions of anthropogenic-driven modification(s) on native ecosystems are worth investigating to enhance understanding of the insect invasion process. In addition, the comparison of native and invasive congeners is recognised as a useful approach for identifying characteristics that promote invasiveness (*Muñoz & Ackerman, 2011*). This approach is perhaps even more useful in this study because the 'invasive congener' is native itself and it would not have been subjected to differential environmental and ecological pressures as its congener that are likely to have affected its evolution.

In New Zealand, the introduction of exotic pastoral plants has resulted in alteration of the diet of the native coleopteran *Costelytra zealandica* (White) (Scarabaeidae), also known as the New Zealand grass grub or brown beetle. The larvae of this endemic insect feed intensively on the roots of ryegrass (*Lolium* spp.) and white clover (*Trifolium repens*) and as a consequence the species is ranked as a major economic pest in New Zealand (*Pottinger, 1975*; *Richards et al., 1997*). Interestingly and in contrast, *C. brunneum* (Broun), a close congeneric species that is rarely found in ryegrass and white clover pastures and remains mostly distributed in native habitats (*Given, 1966*; *Lefort et al., 2012*; *Lefort et al., 2013*). Both *Costelytra* species are considered to be univoltine organisms (*Atkinson & Slay, 1994*) with three larval stages, although it is not uncommon to come across individuals that follow a two-year life cycle in the highest and coldest environments of the southern locations of New Zealand, such as Otago and Southland (*Stewart, 1972*; *Kain, 1975*). These two species are sympatric and share similar native hosts, mainly comprising tussock species (Poaceae) commonly found in New Zealand native grasslands (*Given, 1966*; *Lefort et al., 2012*; *Lefort et al., 2013*).

The present study aimed to investigate the feeding preferences and fitness response in terms of survival and weight gain of these two coleopteran species, to provide new insights into the mechanisms underpinning the invasion process in *C. zealandica*. The first objective of this study was to perform choice tests where the larvae of both *Costelytra* species were given the choice between a native and an exotic host plant. The second objective was to compare survival and larval growth of two populations of the invasive species *C. zealandica* when exposed to these host plants.

## MATERIAL AND METHODS

### Insect sampling and plant material

Newly hatched third instar larva, as the most damaging life stage of the invasive species *C. zealandica* and the most intensively feeding life stage in *Costelytra* spp. in general, were used for the experiments. No protocol exists to rear *Costelytra* spp. offspring under laboratory conditions and all attempts to do so have been unsuccessful. Therefore, the second best option was to work with field-collected insects. Four sampling sites in the

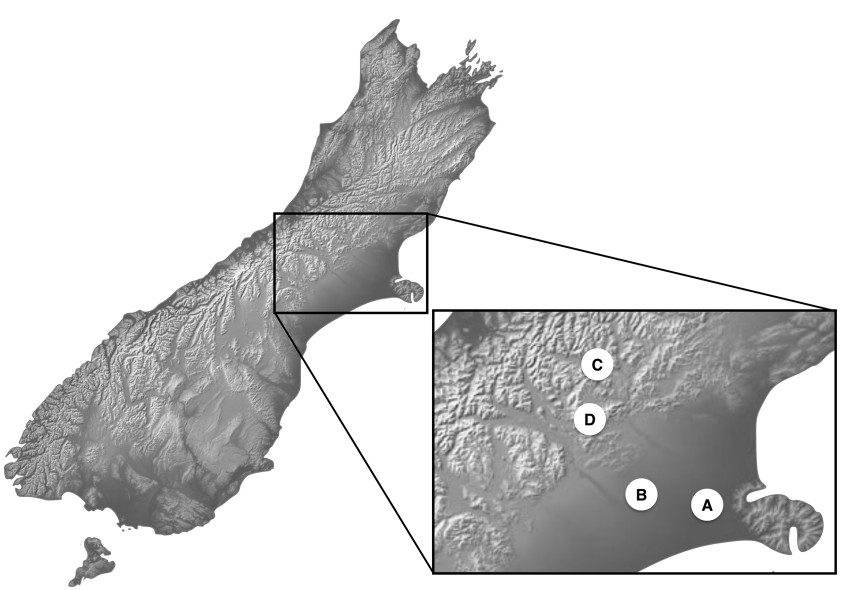

**Figure 1  Location map for *Costelytra zealandica* and *C. brunneum* sample sites.**

**Table 1  General description and location for *Costelytra zealandica* and *C. brunneum* sample sites.**

| Site | Location | Coordinates | Site description and dominant group of plants | Species sampled and population indexing |
|------|----------|-------------|-----------------------------------------------|------------------------------------------|
| A | Lincoln (NZ, South Island) | 43°64′04″S 172°47′82″E | Mixed exotic ryegrass (*Lolium* spp.)/ clover garden (*Trifolium* spp.) | *Costelytra zealandica* (population A) |
| B | Hororata (NZ, South Island) | 43°32′17″S 171°57′16″E | Mixed exotic ryegrass (*Lolium* spp.)/ clover dairy pasture (*Trifolium* spp.) | *Costelytra zealandica* (population B) |
| C | Cass (NZ, South Island) | 43°02′10″S 171°45′40″E | Native tussock grassland (*Poa cita* over 80% incidence) | *Costelytra zealandica* (population C) |
| D | Castle Hill (NZ, South Island) | 43°12′20″S 171°42′16″E | Native tussock grassland (*Poa cita* over 80% incidence) close to the margin of beech forest (*Nothofagus* spp.) | *Costelytra brunneum* |

South Island of New Zealand were used to collect second instar larvae of *Costelytra* spp. (Fig. 1). These sites are labelled A, B, C and D in Table 1. Collection sites A and B were dominated by exotic plants, while sites C and D were essentially composed of native grasses (Table 1). In the two latter sites, larvae of both species were collected under large patches of native vegetation. These patches were distant enough from exotic vegetation, to ensure that no- or minimal-contact with exotic plants had occurred prior to experiments, given the very low mobility of the earliest larval stages in *Costelytra* spp (*Kain, 1975*).

Initially, the larvae were placed individually into ice tray compartments with a piece of carrot as food at 15 °C ambient temperature for four days to test for the presence of the endemic amber disease (*Serratia* spp.) according to the protocol of *Jackson, Huger & Glare (1993)*. Healthy larvae were identified to the species level based on the non-invasive methodologies developed by *Lefort et al. (2012)* and *Lefort et al. (2013)*.

*Trifolium repens* (white clover) was grown in a glasshouse (Lincoln University, New Zealand) from seeds (PGG Wrightson Seeds Ltd, Christchurch, New Zealand) in 200 ml of potting mix comprising 60% peat and 40% sterilized pumice stones. Young plants of the native *Poa cita* (silver tussock) were purchased from a native plant nursery in Christchurch, New Zealand. Each plant was carefully transferred from its original pot to a 200 ml pot, filled with potting mix as described above, and was allowed to grow for 2 months prior to the feeding experiment.

### *Costelytra* spp. feeding preferences—native vs exotic host choice test

The feeding preferences of *C. zealandica* and *C. brunneum* larvae were tested using a three choice olfactometer with native or exotic hosts at 15 °C. The olfactometer comprised of three extended arms, each 120 mm in length and 40 mm in diameter, filled with gamma-irradiated soil (Schering-Plough Animal Health, Wellington, New Zealand) and a $40 \times 40$ mm central exposure chamber. The larvae were introduced through an aperture in the central chamber. A pot containing either no plant (control pot), white clover, or silver tussock was connected at the end of each arm. Third instar larvae of *C. zealandica* collected from sites B (exotic pasture, $n = 35$) and C (native grasslands, $n = 35$) and *C. brunneum* from collection site D (native grasslands, $n = 35$) were used for this experiment. For each population, the bioassay was replicated seven times, with five new larvae inserted together in the central exposure chamber, in order to mimic the natural clustered distribution of the larvae in the field and to test a greater number of larvae. After 24 h, pots were disconnected from the olfactometer device, emptied of their contents and larvae were counted. Between each trial, all components of the olfactometer were washed thoroughly with warm water and left to soak in clean water overnight, finally being left to air-dry on a clean counter and reassembled. Results were analyzed with GLMs (family = Poisson) using R software (*R Core Team, 2014*). Two separate GLMs were performed: (1) choice (plant) vs no choice (control or no choice) and (2) native host plant vs exotic host plant as response variables, a subset of the choice data. The populations of *C. zealandica* and *C. brunneum* from the different sites were analyzed separately.

### *Costelytra zealandica* fitness response on different host plants

Newly moulted third instar larvae of *C. zealandica* collected from sites A (exotic pasture, $n = 64$) and C (native grasslands, $n = 47$) were randomly allocated to the two different host plant treatments (white clover and silver tussock). Each larva was kept individually in a 35 ml plastic container containing 50 g of gamma-irradiated soil (as above) and was fed *ad libitum* with roots of white clover or silver tussock. Containers were randomly arranged on plastic trays and kept in an incubator at 15 °C.

The fresh weight of each larva was recorded at the beginning of the experiment and after the first six weeks of treatment. The latter corresponded to the most intensive weeks of feeding for the third instar life stage of this species. All measurements were performed on a 0.01 g readability portable digital scale. The experiment was conducted over an additional

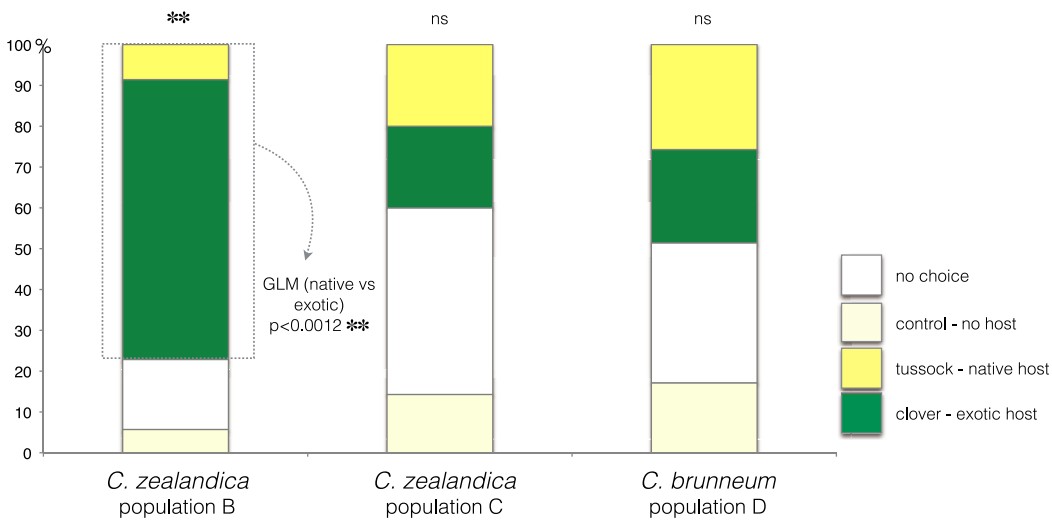

**Figure 2 Plant choice of larvae of three populations of *Costelytra* in a three-arm olfactometer.** With choices of *C. zealandica* population B collected from exotic pastures, *C. zealandica* population C collected from native tussock grassland, and *C. brunneum* population D collected from native tussock grassland. ** indicates $p < 0.01$ and ns indicates $p > 0.05$.

9 weeks, to cover the average 15 week duration of the third instar in *C. zealandica*. Survival rates were assessed after this time.

Statistical analyses to determine the effect of host plant diet on larval survival were carried out using a Chi-squared test. For each population, an ANCOVA was conducted to analyze the effect of host plant diet on larval growth after 6 weeks while controlling for initial weight. The analyses were performed after exclusion of larvae that died before the end of the sixth week. The Chi-squared test was conducted using R software (*R Core Team, 2014*), while the ANCOVA were performed using the statistical software SPSS v.20.

# RESULTS

## *Costelytra* spp. feeding preferences—native vs exotic host choice test

In the choice test, only *C. zealandica* collected from exotic pastures (population B) showed a preference for the exotic white clover (GLM, $p < 0.01$, Null deviance $= 15.04$, Residual deviance $= 4.15$) (Fig. 2). In contrast, *C. zealandica* collected from native grassland (population C) and *C. brunneum* (population D), did not show a preference for either plant species (respectively: GLM, $p = 0.24$, Null deviance $= 23.33$, Residual deviance $= 21.92$, and GLM, $p = 0.87$, Null deviance $= 8.31$, Residual deviance $= 8.28$) (Fig. 2).

## *Costelytra zealandica*—larval survival and growth on exotic clover or native tussock

The larvae collected from exotic pastures (population A) displayed survival rates over six time higher when fed on clover (33.3% survival) compared with larvae fed on native silver tussock (5.5% survival) ($\chi^2 = 4.43, df = 1, p < 0.05$) (Fig. 3).

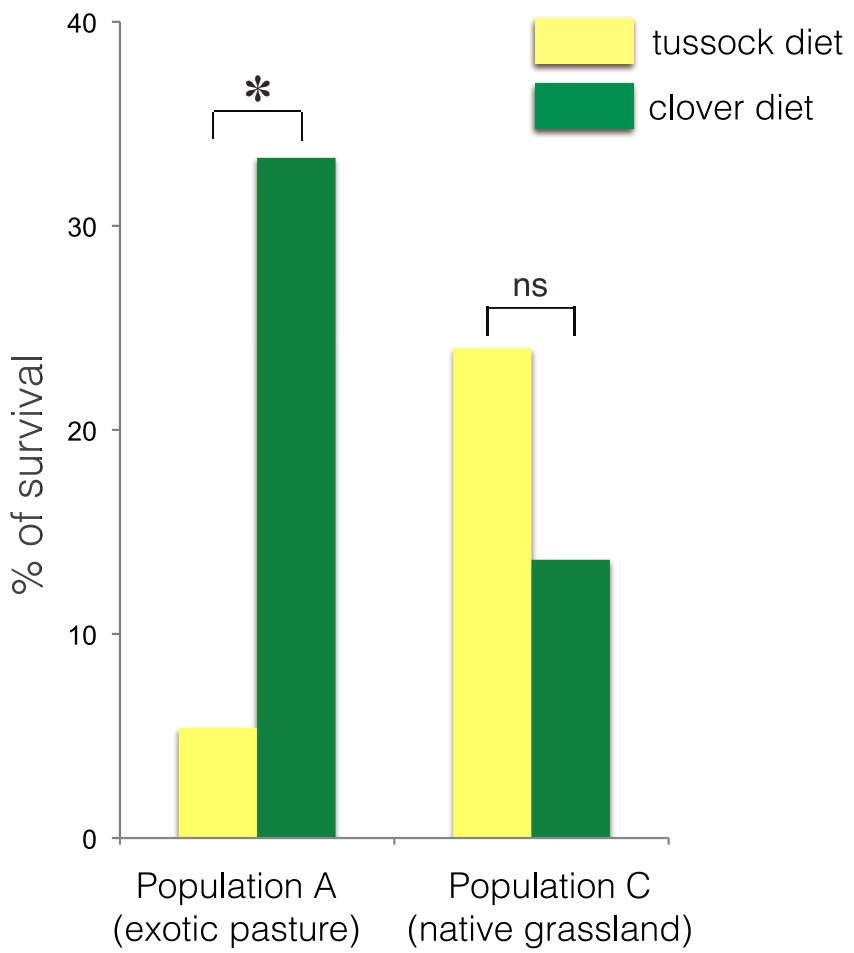

**Figure 3** Percentage of larval survival of *Costelytra zealandica* from site A (collected from exotic pasture) and site C (collected from native tussock grassland) after 15 weeks of feeding on tussock (yellow bars) and white clover (green bars) host plants. * indicates $p < 0.05$ and ns indicates $p > 0.05$.

No treatment effect on larval growth was detected for the population from native grasslands (population C) ($F(2, 22) = 3.69$, $p = 0.07$) (Table 2), while the larvae from exotic pastures (population A) gained 0.428 g ($\pm 0.005$ g) when fed on clover for 6 weeks, which was almost twice as much weight compared with larvae fed on native tussock ($F(2, 54) = 12.26$, $p < 0.001$) (Table 2), (Fig. 4).

## DISCUSSION

This study investigated variation in feeding preferences and fitness response to various hosts in *C. zealandica*. The results corroborate the existence of strong intraspecific variation of the diet breadth of this pest species (*Lefort et al., 2014*). This study also demonstrated similarities between feeding preferences of a population of *C. zealandica* collected from an isolated native habitat with those of the congeneric non-pest species *C. brunneum*. The overall results of this study have provided new insight into the mechanism(s) underpinning the invasion of *C. zealandica* into improved pastures throughout New Zealand.

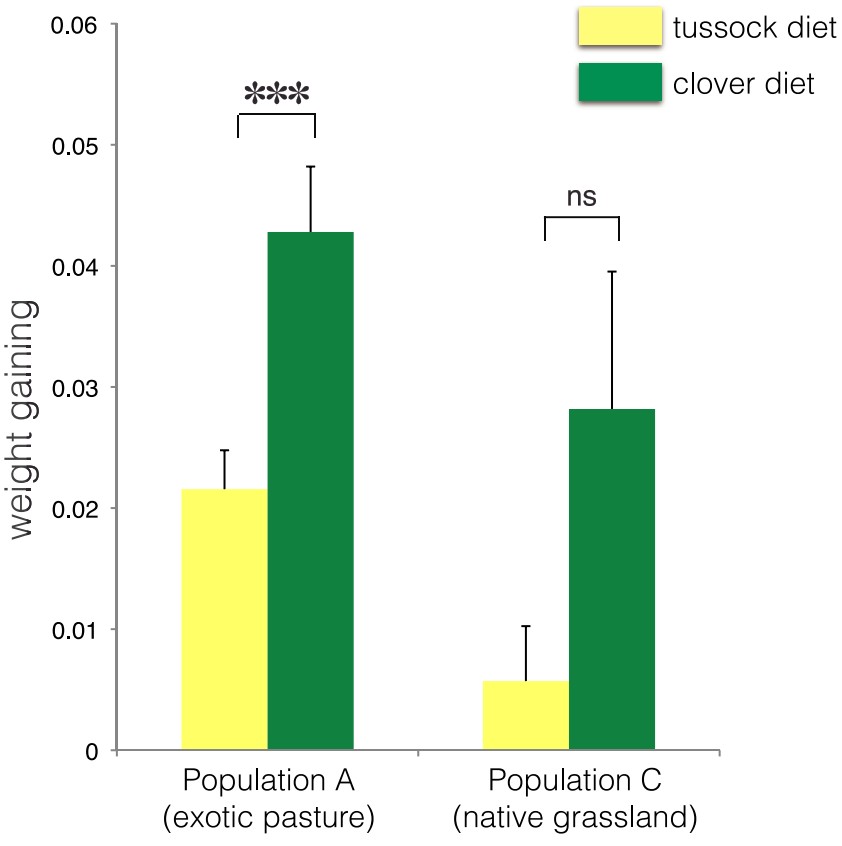

**Figure 4  Average fresh weight gain (+1SE) of larvae of *Costelytra zealandica* from site A (collected from exotic pasture) and site C (collected from native tussock grassland) after 6 weeks of feeding on tussock (yellow bars) and clover (green bars) host plants.** Pairwise comparisons were performed using an ANCOVA with the initial weight of the larvae as covariate. *** indicates $p < 0.001$ and ns indicates $p > 0.05$.

**Table 2  ANCOVA—effect of different host plant diet on the average weight gain of *Costelytra zealandica* larvae controlling for their initial weight.**

| Species (sampling site) | df | F | P values | 5% significance level |
|---|---|---|---|---|
| *C. zealandica* (population A) | | | | |
| Treatment | 1 | 12.257 | 0.001 | *** |
| Covariate (initial weight) | 1 | 0.001 | 0.978 | ns |
| Error | 54 | | | |
| *C. zealandica* (population C) | | | | |
| Treatment | 1 | 3.691 | 0.068 | ns |
| Covariate (initial weight) | 1 | 0.190 | 0.667 | ns |
| Error | 22 | | | |
It is important to note that the nutritional value of the roots on which the larvae fed can vary within the same plant in response to soil nutrient distribution and concentration (*Grossman & Rice, 2012*) and possibly results in differential fitness performance in the same population of insect. However, the overall fitness, as measured by survival and growth, of *C. zealandica* collected from exotic pastures was significantly higher on the exotic host plant than on its native host. Inheritance and maternal effects on host choice (*Mousseau & Dingle, 1991*; *Mousseau & Fox, 1998*), where offspring display high fitness performance (*Fox & Wolf, 2006*) and similar host preferences to their mother (*Craig, Horner & Itami, 2001*), is a possible explanation. Similarly, another maternal effect coined the 'mother knows best' hypothesis, which suggests that females tend to oviposit on host plant(s) that can potentially increase their offspring survival (*Scheir, De Bruyn & Verhagen, 2000*; *Mayhew, 2001*), can also be a possible explanation, although no evidence supporting this hypothesis has been observed in *C. zealandica* adult beetles (*Kelsey, 1968*; *Radcliffe & Payne, 1969*; *Kain, 1975*).

The effects described above are supported by the results of the choice test. In this test, population A, consisting of *C. zealandica* larvae collected from exotic pasture plants on which the population is likely to have fed for several generations, chose exotic clover as the preferred host plant. In contrast, the population of *C. zealandica* collected from their native range did not show any preference in the choice tests and did not perform better on either host. The first observation negates the hypothesis of inheritance and maternal effect on host choice mentioned earlier, since based on this principle, this population would have been expected to prefer its native host (i.e., silver tussock) and have better fitness performance on this plant compared with the exotic host (i.e., white clover). Unlike silver tussock, white clover is a legume, which may partly explain the differences in larval weight gain observed in the *C. zealandica* population collected from exotic pastures. Indeed, because of their bacterial symbiosis resulting in an ability to fix nitrogen (*Awmack & Leather, 2002*), the nutritional value of this family of plants is likely to be higher than that of grasses, such as silver tussock, used as the alternative host in this study. However, this alternative hypothesis does not explain the response of the other *C. zealandica* population studied, which in this case would have been expected to show increased weight gain on clover as well.

Based on similar survival rates observed in the two populations of *C. zealandica* used in this study, and because the population collected from native grassland was presumably isolated enough to have not fed on exotic host plants prior to the experiment, it appears that the successful exploitation of an exotic plant by this species is likely a pre-existing ability. *Diegisser et al. (2009)* and *Ding & Blossey (2009)* suggested that some form of pre-adaptation was required for the exploitation of a novel host plant. The hypothesis of pre-adaptation or phenotypic plasticity in *C. zealandica* is supported by (i) the similarity in host choice between larvae of *C. zealandica* collected from native grassland and larvae of the non-pest species *C. brunneum*, and (ii) the differential exploitation of exotic pastoral plants by the two species. However, the limited number of replicates for the population collected from native grassland calls for caution in the interpretation of these results.

The defence mechanisms employed by the different host plants and their effect on the fitness of the insect species studied would be an interesting aspect to investigate. In a recent review about phytophagous insects and plant defences, *Ali & Agrawal (2012)* reaffirmed that generalist insects do not master or totally overcome their host defences, but possess 'general mechanisms' to tolerate an array of those defences. It is possible to observe variations in this tolerance, particularly when the host-range utilised by the insect species is highly diversified and, consequently, when the family of plants have differential evolutionary histories that may have resulted in slight variations in their defence mechanisms. Here, *C. zealandica* may have been less affected by the defences of white clover compared to those of the other hosts or, conversely and as recently shown by *Lefort et al. (2015)*, may have benefited from the defences of their host. The latter phenomenon has been observed several times in recent insect-host interaction studies, where the defences of the hosts were artificially triggered and the resulting fitness response of the insects studied unexpectedly enhanced (e.g., *Pierre et al., 2012*; *Robert et al., 2012*).

The results of this study support the pre-existence of characteristics that may have contributed to the invasion success of the New Zealand native scarab *C. zealandica* into exotic pastures throughout New Zealand in contrast to its native congener, *C. brunneum* which maintains small populations in native grasslands. Additionally, the differences in feeding preferences between different populations of the pest species *C. zealandica*, seem to confirm recent evidence (*Lefort et al., 2014*) of the existence of distinct host-races in this species. With regard to cryptic species, many studies have highlighted the importance of correct species identification for the accomplishment of successful biological control (e.g., *Rosen, 1986*; *Paterson, 1991*; *Silva-Brandão et al., 2013*). Similarly, we believe that the delineation of host-races in pest species could have vital implications in terms of pest control management and strategies. For instance, caution should be taken before denominating a species as a single entity by employing terms such as "pest species" or "invasive species," and care must be taken during insect sampling and subsequent identification, particularly when performing bioassays for which the outcome may vary depending on the host-race used. Because the natural feeding behavior of some insects can be modified by laboratory experimentation, we believe that complementary *in-situ* experiments that would allow the incorporation and investigation of the effect of natural environmental variables on the feeding behavior of *C. zealandica*, would be beneficial. Furthermore we strongly encourage further molecular investigations to confirm the possible existence of host-races in *C. zealandica*, which would greatly benefit the field of biological control research in New Zealand.

## ACKNOWLEDGEMENTS

The authors would like to thank Richard Townsend and St Andrew's College of Christchurch for granting access to the different insect collection sites.

### Funding

Financial support was provided by Miss EL Hellaby Indigenous Grasslands Research Trust, Better Border Biosecurity and the Bio-Protection Research Centre. The funders had no role in study design, data collection and analysis, decision to publish, or preparation of the manuscript.

### Grant Disclosures

The following grant information was disclosed by the authors:
EL Hellaby Indigenous Grasslands Research Trust.
Better Border Biosecurity and the Bio-Protection Research Centre.

### Competing Interests

The authors declare there are no competing interests.

### Author Contributions

- Marie-Caroline Lefort conceived and designed the experiments, performed the experiments, analyzed the data, wrote the paper, prepared figures and/or tables, reviewed drafts of the paper.
- Stéphane Boyer conceived and designed the experiments, performed the experiments, reviewed drafts of the paper.
- Jessica Vereijssen conceived and designed the experiments, reviewed drafts of the paper.
- Rowan Sprague analyzed the data, reviewed drafts of the paper.
- Travis R. Glare and Susan P. Worner contributed reagents/materials/analysis tools, reviewed drafts of the paper.

### Data Availability

The raw dataset has been made available online Data S1.

### Supplemental Information

Supplemental information for this article can be found online at http://dx.doi.org/10.7717/peerj.1454#supplemental-information.

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
