# Peer review of "Preference of a native beetle for “exoticism,” characteristics that contribute to invasive success of Costelytra zealandica (Scarabaeidae: Melolonthinae)"

_PeerJ, doi:10.7717/peerj.1454_

## Round 0.1 · original submission · Major Revisions

All three reviewers have made substantive editorial comments on your manuscript, which I believe are valid and thorough. Please address these comments in your revisions, particularly in regards to re-analysis of your data and associated figures, and inclusions of tables to make the information easier to follow.

·

Basic reporting

• Line 27: For “feeding preference” and “fitness performance”, some idea of the variables measured should be used, especially as feeding preference and actual fitness were not measured, rather surrogate variables. Something like “(olfactometer choice)” and “(third instar weight increase)” would suffice
• Lines 33-37: I think the results part of the abstract could be stated more clearly and completely. Information is missing regarding the preference experiment. I would modify to something like: “Costelytra zealandica, when sampled from exotic pastures, showed higher preference and performance on exotic plants. In contrast, C. zealandica sampled from native grasslands did not perform significantly better on either host and showed similar feeding preferences to C. brunneum, which exhibited no preference.”
• Line 79: Like in the abstract, I think the terms feeding preference and fitness response need to be more explicitly outlined by mentioning the actual variables measured.
• In general, more detailed and interesting information could be provided in the introduction, especially as the paper is relatively short. Broadly, a greater discussion of the background literature informing the theory/hypotheses the objectives are based on would be useful – consider including your predictions based on the theoretical expectations. In a more narrow sense, quite a lot is known about the study species, so this information should be included with a bit more detail, perhaps with a particular focus on the authors’ prior work (i.e., Lefort et al. 2014, 2015, etc.) and basic biology/ecology (i.e., relationship with pathogens or other microbes)
• Lines 89-90: Can you provide references here? This entire sentence is rather clumsy, consider rewriting
• A simple table summarizing the details of each collection site/species combination would be useful, appropriate, and would lead to very quick understanding of the experimental design and main results. Consider including: 1) location, 2) dominant plant species at each site, 3) Costelytra species collected, 4) feeding preference, 5) survival rate and 6) mean weight increase on each different food source
• Lines 104-105: “given the very low mobility of the earliest larval stages in Costelytra spp.” – references must be provided for statements such as this
• Line 163: Give all appropriate statistical results, including ones which show no difference/preference
• Lines 166-169: What were the survival rates for the larvae from population C (native grasslands)? Report these. Provide the statistical results for population C. Also, did the overall survival and performance of larvae differ between sites A and C? As mentioned in the methods section, a two-way ANOVA may be a more appropriate analysis than a t-test
• Lines 171-175: Present means, standard errors, and effect sizes to go along with the stats
• Can the C. zealandica host races be identified genetically or morphologically?
• Using colours in the figures makes things difficult for people who may be colourblind – try using simple patterning or shading for figures 2-4
• Figure 2: Further to my previous point, it is very difficult to distinguish the different treatments in the pie charts. There appears to be a transition of some sort between native and invasive choices, rather than showing the exact proportions.
• Figures 3 and 4: Ideally, the figures would be self-contained and able to be fully interpreted without the caption. Hence, the x-axis label should reflect if the source population was native or invasive, and bar colour should be included as a figure key/legend.
• Figure 3, Line 343: In the methods and results, measurements from week 6 were used. Both sets of data should be included in the analyses and presented in the paper.
• Figure 3: 95% confidence intervals could be bootstrapped to use as standard error bars

Experimental design

• Line 111: Given temperature is important for Costelytra spp. development and fitness, what were the average environmental conditions in the glasshouse?
• Lines 121-126: This is hard to visualise. A diagram showing how the olfactometer was set up may be useful
• Line 130: What is the rationale for using 5 larvae at a time? Could this have had an effect on the results?
• Lines 142-147: Is it hard to grow Costelytra spp. in the field or in pots? Replicating the experiment in this way would be more similar to field conditions. Alternatively, could you weigh larvae collected at the end of the growing season from areas dominated by native and exotic hosts? This would make the study more robust and allow for greater inference, given the results are no longer restricted to a simplified laboratory experiment
• Lines 142-147: Why was the experiment not also replicated with C. brunneum? It would help demonstrate any differences in C. brunneum development on native or exotic species. Could additional Costelytra species also be used (see Given 1966)? Some further explanation or justification should be given
• Lines 150-152: Was the weight also measured at the end of 15 weeks, which is the usual length of the third instar as mentioned by the authors? It would seem this final fitness measurement (or even dry weight of adults) is more appropriate, especially given this was when larvae survival was quantified. If this data was recorded, then it should be included in the analysis and results. If not, rationale should be provided in the paper as to why it was not measured or used. In its current form, it is very confusing as to what data was used for the analysis
• Line 152: Using a balance? To what level of error?
• Line 154: How was average larval growth calculated? Include equation
• Line 155: Looking at raw data, it seems there are a number of larvae which survived but did not increase in weight at all (some even decreased), implying they had completed development upon collection. What would be the biological rationale for this data to be included? This should be pointed out in the methods section or this data also removed from the analyses. Finally, I’m a little concerned that measuring fresh weight is not the same as measuring dry biomass, as fresh weight can be dependent on water content at the time of weighing. Can the authors alleviate this major concern in some way, such as weighing adult beetles reared on different species, or discussing these limitations in the paper? Furthermore, the raw data should be modified to include a column showing the weight increase

Validity of the findings

• Line 135: Is a chi-squared test the most appropriate analysis? I believe a better analysis may be using two separate general linear mixed models with 1) choice (plant) vs no choice (control or no choice) and then 2) native vs exotic used as binomial response variables, tested against Costelytra species and source population, with replicate added as a random effect
• Lines 153-154: Again, is a chi-squared test the most appropriate analysis? I believe a better analysis may be a general linear model using a binomial error structure and including source population and diet treatment as interacting fixed factors. Furthermore, a two-way ANOVA for larval growth would also be more appropriate – a significant interaction would indicate the same result/implications as your separate t-tests, while also controlling for site of origin
• Data has been made available with submission.
• Conclusions are appropriate but be careful with some implications; the results are not replicated in the field or pots where C. zealandica may feed in a more natural setting. These limitations should be mentioned as a caveat somewhere in the discussion
• Perhaps the authors can expand upon the potential implications of the research a little more. The whole host-race or genotype point is particularly interesting and worthy of further discussion; other species (e.g., Phragmites australis) have shown similar patterns of having invasive and non-invasive genotypes

Additional comments

This interesting study addresses mechanisms underlying invasion success of Costelytra zealandica, a native beetle species which spread extensively and became a pest following the introduction of exotic crop species such as ryegrass and clover. The authors collected third instar larvae of C. zealandica and C. brunneum from areas of native or exotic plants from four field sites to conduct two experiments: 1) an olfactometer choice test (silver tussock, white clover, control, no choice), and 2) a fitness experiment where C. zealandica from native and exotic sites was fed roots of either its native or exotic host species.

In the olfactometer experiment, the authors found that C. zealandica collected from exotic pasture showed a preference for the tube containing the exotic species, whereas C. zealandica and C. brunneum collected from under native plants showed no preference for either native or exotic plants. The second experiment showed that larvae collected from exotic pasture survived and performed better when fed white clover roots than roots of native silver tussock, whereas larvae collected from native sites demonstrated no difference in larval weight increase when fed native or exotic plant roots. These results indicate that there may be distinct host-races of C. zealandica which prefer and perform more strongly on exotic crops, underlying why the species has become invasive on these crops.

Overall, I like the paper. It is clear that a lot of time and hard work has gone into collecting the beetle larvae and running the experiments, which I certainly appreciate. Furthermore, I believe the study is scientifically sound, reasonably well-written, and falls under the broad scope of the journal. Indeed, the use of a native species as an invader alongside a closely-related benign native species is an excellent phylogenetic control, as well as adding to the currently small body of literature on native invaders. Furthermore, evidence for distinct host-races of C. zealandica is an interesting and important result and so should be published. However, I do have a few issues with certain parts of the paper, particularly with completeness of data and results, the statistical approach, and some issues with experimental design (see specific comments). While these revisions will take some time and could be regarded as major, once rectified they will make the paper more thorough and of higher quality. Minor general comments and corrections are as follows:
Title
• Line 15: I would suggest using a colon instead of a fullstop
• Line 16: Try using “of” instead of “in”
Abstract
• Line 23: “resulted in diet alteration”
• Line 26: Consider replacing “new” with “novel” and remove the quotation marks
• Line 30: “To this end,” and “third instar larvae”
• Line 32: Replace “were” with “was”
• Line 38: “Intraspecific” and no comma following “variation”
Keywords
• Consider adding “grass grub” to keywords
Introduction
• Line 55: Consider using novel instead of new, and remove the quotation marks.
• Lines 56-57: “Some native species.”
• Line 61: “modifications to”
• Line 64: Why is a native invader “even more so” useful than an introduced invader? Provide a rationale or remove.
• Line 67: Give common name also, especially seeing as it is a well-known species
• Lines 67-69: The sentence is a bit clumsy, try rewriting it
• Lines 68-69: Give the scientific names for these crop species
• Lines 69-70: The formatting of the references is inconsistent. Here a comma is used before the date of publication, where previously no comma was included. Please check the manuscript for formatting consistency.
• Line 73: “are considered to be univoltine (Atkinson…”
• Line 81: This is a little broad, modify to just “the invasion process of C. zealanadica.”
Materials and Methods
• Line 91: “used for the experiments”
• Line 91: Replace “produce” with “rear”
• Lines 92-93: “the second best option”
• Line 94: “the South Island of New Zealand”
• Line 94: Use a colon instead of a comma, i.e. “Costelytra spp.:”
• Lines 95-97: Try “Lincoln (43°64’04’’S 172°47’82’’E), Hororata (43°32’17”S 171°57’16”E), Cass (43°02’10”S 171°45’40”E) and Castle Hill (43°12’20”S 171°42’16”E), as sites A, B, C and D, respectively (Figure 1).”
• Line 101: Remove the word “being”
• Line 103: Remove comma
• Line 107: Insert a space into 15 °C
• Line 108: Give scientific name (Serratia spp.)
• Line 120: “preferences”
• Line 127: Try changing to something more concise like “populations B (exotic pasture, n = 35) and C (native grasslands, n = 35)
• Line 132: “contents”
• Line 138: “post-hoc”
• Line 145: Be consistent with spacing between numbers and units, i.e., “50 g” and “n = 64”
• Line 147: Insert a space into 15 °C
• Line 154: “Welch’s t-test”
• Line 154: delete extra space between “t-test for”
Results
• Line 161: Give actual p-value unless < 0.001
• Lines 162-163: Try “C. zealandica (population C) and C. brunneum (Population D) collected from native grassland did not show a preference for either plant species”
• Line 165: Use the same header as in the methods section
• Line 169: Give actual p-value unless < 0.001
Discussion
• Line 178: “variation”
• Line 179: Try “existence of strong intraspecific variation in diet breadth”
• Line 181: Delete “the”
• Lines 183-185: “such as …”? Be specific about the insights.
• Line 187: Higher than? Be specific and don’t leave anything up to interpretation
• Line 191: “’mother knows best hypothesis’”
• Line 193: “could also be a possible explanation, although…”
• Line 193-194: “evidence supporting this hypothesis”
• Line 196-199: This is a clumsy sentence, try rewriting
• Lines 199-200: Try “collected from native grassland species”
• Line 213: This long paragraph can probably be broken in two when you switch to talking about defences
• Lines 222-224: Not clear what the question is here, try rewriting this sentence
• Lines 225-226: “do not master or totally overcome…”
• Line 230: “resulted”
• Line 230-231: Try something more concise and clear like “Here, fitness of C. zealandica may have been less affected by the defences of white clover compared to those of the other hosts, or recent evidence suggests larvae may also benefit from the defences of their host (see Lefort et al. 2015)”
• Line 240: Replace “that” with “which”
• Line 241: Delete comma
References
• Line 259: Fullstop at end of reference. Check all references for consistency
• Line 260: Species name should be italicized
• Line 268: Inconsistent formatting (capitalization)
• Line 277: Insert Costelytra in appropriate place
• Line 279: “Glare TR, 1993” – check references throughout manuscript
• Line 281 and line 309: Species name should be italicized
Figures, Tables, and Captions
• Figure 2: Sub-figures are not labelled (a), (b), or (c) in the figure itself
• Figure 2, Line 337: Delete extra space between “of (a)”
• Figure 3, Line 342: Delete extra space between “zealandica from”
• Figure 3, Line 343: Delete extra space between “feeding on”
• Figure 4: Error bars should extend above and below mean on these bar plots
• Figure 4, line 350: “Welch’s t-test”

Reviewer 2 ·

Basic reporting

The paper is generally well written and meets the standards required of the journal. There are some odd usages in the text (especially the title) and a couple of typographic errors (e.g. extra word in line 86, use of hyphens in line98) but overall it is sound. I don’t think the use of the word “fond” in the title is appropriate, it is too anthropomorphic. I think it should be changed.
I found the pie charts in Figure 2 to be confusing. I feel it would be more useful to present these as stacked bar graphs. They statistics attached to those figures are also too precise. I think Figs 3 and 4 may also need to be modified to accommodate a different analysis.

Experimental design

The choice experiments are interesting and the use of the 3-armed olfactometer for these experiments seems to be similar enough to how larvae find food in the field. Are larvae normally found together? The decision to use groups of 5 larvae in the trials needs to be justified as the patterns observed may also be a function of a single larva’s choice rather than each individual choosing the ultimate host.
Some of the analyses could be improved. Could survivorship curves with Meir-Kaplan analyses be performed with the data obtained? That would be more informative than just using the final mortality rates.
The analysis of the feeding trials should also be improved. Using the raw mass gain isn’t really appropriate to compare performance on the different diets, it would be better to use something like ANCOVA using initial size as a covariate.

Validity of the findings

The data collected are interesting but the statistics used to analyse them are not the best options for these data. The work still seems to show that the two species have a different affinity for different host plants and how this is reflected in host choice and use. The speculation on host race bit seems a bit out of place here but is grounded in the paper (and the work associated with it) so it’s reasonable to include it.

Additional comments

This is a really interesting system and the questions raised in the paper are of broad significance to a number of biologists. It’s difficult to put the host shifts in the context of how the two species use different hosts naturally, as little is known of the their breadth in unmodified environments (short of the comments in line 73) and this is something that would really advance the questions raised here. I think the work would be made much better with some reanalyses and some rejigging of how the data are presented.

Reviewer 3 ·

Basic reporting

No Comments

Experimental design

The experimental design is appropriate for the stated objectives. It might have been more appropriate to make the comparison between C. zealandica populations feeding on native and exotic plant species if they had chosen individuals feeding on ryegrass as an exotic instead of white clover. In that way they would be comparing fitness of the populations on a native grass and an exotic grass rather than between a grass and a legume with all the attendant symbiotic associations that could affect the results.

Validity of the findings

The conclusions leave out a possible alternative explanation that should be addressed. That is, it may be possible that the authors are dealing with two different cryptic species when comparing individuals from two different populations. I am not necessarily suggesting that they conduct a genetic analysis, but I do think they need to address this possibility, particularly since they do raise the issue of the exotic plants being a selective force in the potential fitness of the herbivore.

Additional comments

I encourage the authors to carefully edit the text to eliminate cases of awkward or convoluted phrasing (e.g., lines 54-59, lines 64-68, lines 217-220)

---

## Round 0.2 · Minor Revisions

Both reviewers agree that the manuscript is much improved. Please address all the grammatical issues raised by Reviewer 1. Also for the For the larval growth bioassay, consider the ancova. analysts.

Figure 2 - a stacked bar graph may be more appropriate here. You should also include the figure title next to the graph letter e.g. 'a) C. zealandica population B,' rather than a separate box. If you keep as a pie chart, please make differentiation between diets more pronounced - they look like they blend in to one another.

·

Basic reporting

• Figure 1 is not referred to in the text. This should be rectified at the start of the methods section.
• The authors must include more information in the results section. Present the means and associated standard errors/confidence intervals. Effect sizes must also be included, e.g., “x% of larvae chose the exotic host, 5 times more than the x% which chose the native host.” I understand some of the data is already presented in tables and figures, but thoroughness when presenting results is of utmost importance and will certainly not detract from the paper.
• The survival results for larvae from population C are not reported at all in the results, which is a comment I also made on the last review. This must be included. While the statistical analysis shows the result to be non-significant (as seen in Fig. 3), there is almost a 100% greater survival rate on native plants. This is a large effect size and should not be ignored. I would state the effect size in the results with the caveat that the stats show it to be non-significant (possibly due to low sample size, which can cause some issues with ch-squared analysis).
• Lines 176-177: Again, effect sizes must be reported. The effect size between the two treatments is actually larger than the effect size for larvae from population A, and the p-value could be termed marginally significant. I think these results warrant further discussion and point to limitations in sample size

Experimental design

• For the larval growth bioassay, I still disagree with the analysis. A two-way ANCOVA would be more appropriate. This would allow the authors to also consider any differences in growth between the two C. zealandica populations.

Validity of the findings

• I believe the authors’ interpretation of the data to be correct and their conclusions valid. A larger sample size may have helped elucidate more differences among the treatments which were marginally significant, so this should be mentioned in the discussion.

Additional comments

The authors did a good job in taking the reviewers’ comments on board and making necessary corrections to the manuscript or providing further explanation/justification when required. In particular, the better justification of the methods used and the inclusion of more robust analyses has resulted in a much improved manuscript. As such, I have no problem with recommending this paper for publication. However, the authors should pay close attention to their sentence structure. It still appears a little convoluted at times and could use another thorough proofread (some examples are given below). A few other minor points which still require the authors’ attention are listed below:

Abstract
• Line 26: “close” is redundant, remove.
• Lines 31-33: This sentence structure is awkward, try “To this end, the feeding
preference of third instar larvae of both Costelytra species was investigated
under controlled conditions using an olfactometer device, …”
• Line 33: Consider removing “under controlled conditions”, this is assumed for abstracts and is sufficiently outlined in the methods section.
• Lines 49-50: Running head should be modified to reflect new title.
Introduction
• Line 71: Delete the redundant second “the”
• Line 71-74: Grammar of this sentence needs to be fixed.
• Line 75: Replace “not often” with “rarely”
• Line 85: “terms”
Materials and Methods
• Line 100: This is where sites A and B also need to be introduced and described as exotic-dominated.
• Line 153: I think this would be an ideal time to state the reasons why larvae were not weighed again after 15 weeks, as the readers will be wondering this without justification.
• Line 156: replace “and by” with “while”.
• Line 158: Chi-squared or chi-square? Both are used in the same paragraph.
Results
• Use a consistent number of decimal places for each statistic throughout the results section.
• Line 167: Insert (population D) after C. brunneum for clarity.
• Line 177: Check the formatting of statistical summaries – generally df are not given within their own brackets).
Discussion
• Lines 190-195: This sentence is too long. Consider breaking it into two separate sentences, first presenting the key result and then discussing the reasons for this result
• Line 239: Remove comma following “been”
Figures, Tables, and Captions
• Figure 2: The caption is unclear as to whether the a, b, or c is referring to the possible choices or the different populations. I would rephrase this part of the caption for clarity.
• Figure 2: I still don’t like this figure a whole lot, especially the pie chart part. The colors of no-choice and control are virtually indistinguishable and must be changed. Also the merging of the green and yellow color in the pie chart is not representative of the actual proportions of choices observed, particularly for population A. This part of the chart could simply be split into the two colors much like the bars beside it. I really think this data could be presented in a clearer way.
• Figure 4: I disagree with the authors’ assertion that error bars should not extend above and below the mean. The error is not simply calculated in one direction, it goes either side of the mean. Thus, this should be presented in the figure to aid in interpretability. The majority of papers I have read over the last decade have used this format. However, I am willing to be convinced otherwise with a solid argument and references.

Reviewer 2 ·

Basic reporting

The revised version is a much better written manuscript and presents a coherent and thoughtful argument supported by the data. I think the new title describes the work better too. The reason I initially suggested stacked bars was to make the comparison among the 3 populations easier to make. I still think they are a much better option seeing they are used in the comparisons alongside the pie charts.

Experimental design

The explanation for using groups of larvae (my main concern) is explained satisfactorily.

Validity of the findings

No comments (additional to the previous report).

Additional comments

My initial impression of the manuscript was positive and that hasn’t changed. I have only commented on my main concerns here, I think the small changes (and those suggested by the other reviewers) have resulted in a better manuscript.

---

## Round 0.3 · accepted · Accept

I believe all issues raised by the reviewers have been dealt with appropriately, and i think the manuscript is now a much stronger one, now ready for publication.